# Dynamic phosphorylation of Hcm1 promotes fitness in chronic stress

Michelle M. Conti[1], Jillian P. Bail[1], Aurelia R. Reynolds[1], Linnea G. Budge[1], Mackenzie J. Flynn[1], Rui Li[1], Lihua Julie Zhu[1,2,3], Jennifer A. Benanti[1]*

1 Department of Molecular, Cell and Cancer Biology, University of Massachusetts Chan Medical School, Worcester, Massachusetts, United States of America, 2 Department of Genomics and Computational Biology, University of Massachusetts Chan Medical School, Worcester, Massachusetts, United States of America, 3 Program in Molecular Medicine, University of Massachusetts Chan Medical School, Worcester, Massachusetts, United States of America

* jennifer.benanti@umassmed.edu

## Abstract

Cell survival depends upon the ability to adapt to changing environments. Environmental stressors trigger an acute stress response program that rewires cell physiology, downregulates proliferation genes and pauses the cell cycle until the cell adapts. After the acute response is resolved, cells resume cycling but at a reduced rate. The importance of cell cycle changes for survival in chronic stress is not clear. Here, we show that dynamic phosphorylation of the yeast cell cycle-regulatory transcription factor Hcm1 is required to maintain fitness in chronic stress. Hcm1 is activated by cyclin dependent kinase (CDK) during S-phase and is inactivated by the phosphatase calcineurin (CN) in response to stressors that signal through increases in cytosolic $Ca^{2+}$. Cells expressing a constitutively active, phosphomimetic Hcm1 mutant exhibit a reduction in fitness in stress, suggesting Hcm1 inactivation promotes survival. However, a comprehensive analysis of Hcm1 phosphomutants revealed that Hcm1 activity is also important to survive stress, and that all mutants with fixed phosphorylation states are less fit in stress. Moreover, our data suggests that pulses of Hcm1 activity are necessary to maximize target gene expression in stress. These findings demonstrate that expression levels of Hcm1 target genes influence fitness in stress and suggest that the dynamic phosphorylation of cell cycle regulators plays a crucial role in promoting survival in stressful environments.

## Author summary

To survive in a constantly changing environment, cells must be able to adjust their essential processes in response to external stressors. One way they do this is by temporarily slowing down growth and division, giving themselves time to adapt. While this immediate response is well studied, much less is known about

**Data availability statement:** All relevant data are within the manuscript and its Supporting information files, except for Phosphosite Scanning sequencing data and RNAseq data. Phosphosite Scanning data is available from the NCBI Sequencing Read Archive under BioProject # PRJNA1117860. RNAseq data files are available from NCBI-GEO with accession number GSE276435.

**Funding:** This work was supported by National Institutes of Health grant R35GM136280 to J.A.B. The funders had no role in study design, data collection and analysis, the decision to publish, or preparation of the manuscript.

**Competing interests:** The authors have declared that no competing interests exist.

how cells manage long-term stress. In this work, we investigated whether yeast cells need to regulate their cell cycle to survive over extended periods of stress. We focused on the cell cycle-regulatory transcription factor Hcm1, which helps control genes needed for cell division. We found that Hcm1 must be dynamically regulated through phosphorylation to help cells survive in stress. If Hcm1 is always active or always inactive, cells become less fit. Instead, cells need pulses of Hcm1 activity to fine-tune gene expression and cope with stress. Our findings highlight the importance of flexible control over cell cycle processes during stress and suggest that similar mechanisms may help human cells survive in challenging conditions, such as during disease or aging.

## Introduction

Cells are continuously exposed to stressors in the environment and must adapt to these challenges to survive and proliferate. Adaptation not only protects healthy cells from death, but conversely, it can promote the development of disease. For instance, cancer cells must adapt to stressful environments when they metastasize to distant sites [1,2], and fungal pathogens rely upon stress response pathways for survival within the host [3]. Despite the importance of this process, the long-term changes that cells must undergo to maintain fitness and proliferation when faced with chronic stress are poorly understood.

The acute stress response, which occurs immediately following exposure to an environmental stressor, is conserved from yeast to humans and includes a downregulation of protein synthesis and an upregulation of stress response genes [4,5]. In addition to these changes that impact cell physiology, cell cycle-regulatory genes are downregulated, and cells undergo a transient cell cycle arrest [6,7]. After cells adapt to the new environment, the acute stress response is resolved and cells resume proliferation in the new environment, albeit at a reduced rate [8–10]. Transient cell cycle arrest during the acute stress response is thought to be crucial to promote long-term survival [8,9,11]. However, the importance of cell cycle changes for survival in chronic stress is not well understood.

Some stressors activate the conserved $Ca^{2+}$-regulated phosphatase calcineurin (CN), which coordinates both the stress response and cell cycle changes [12,13]. CN activation leads to a decrease in expression of cell cycle-regulatory genes and controls the length of cell cycle arrest, in combination with the stress-activated MAPK Hog1/p38 [14]. One direct target of CN in budding yeast is the S-phase transcription factor (TF) Hcm1 [15]. Hcm1 is a forkhead family transcriptional activator which, like its human homolog FoxM1, plays a crucial role in maintaining genome stability [16,17]. Hcm1 controls expression of key cell cycle genes including histone genes, downstream cell cycle-regulatory TFs, and genes that regulate mitotic spindle function. As cells progress through the cell cycle, Hcm1 is activated by multisite phosphorylation by cyclin dependent kinase (CDK) [18]. CDK phosphorylates eight sites in the Hcm1 transactivation domain (TAD) to stimulate its activity and a three site

phosphodegron in the N-terminus to trigger proteasomal degradation. Immediately following exposure to $CaCl_2$ or LiCl stress, two stressors that activate CN, CDK activity decreases, the activating phosphates on Hcm1 are removed by CN, and expression of Hcm1 target genes decreases [14,15,19]. Whether or not Hcm1 inactivation is critical for cells to adapt and survive in the face of chronic stress is unknown.

Mutations within the Hcm1 TAD have been used to study the consequences of phosphorylation. Phosphomimetic mutations at all CDK phosphosites in the TAD generate a constitutively active protein that leads to increased expression of Hcm1 target genes [18]. In normal growth conditions this mutant provides a fitness advantage to cells, rendering them more fit than wild type (WT) [15,20]. This is a surprising result because advantageous mutations are expected to be selected for during evolution. However, the fact that activating phosphates are removed by CN when cells are faced with stress suggests that dephosphorylation and inactivation of Hcm1 may be necessary for cells to survive in stressful environments.

Here, we investigated this possibility and found that cells expressing a constitutively active, phosphomimetic Hcm1 mutant lose their fitness advantage when they proliferate in LiCl stress for up to 30 generations. To determine the optimal level of Hcm1 activity for fitness in this environment, we screened a collection of Hcm1 mutants that encompass all possible combinations of non-phosphorylatable and phosphomimetic mutations in the TAD, representing the entire spectrum of possible activity levels. Surprisingly, this screen revealed that almost all mutants were less fit, relative to WT, when growing in stress compared to stress-free conditions. Moreover, Cks1-priming sites that stimulate Hcm1 activity by promoting phosphorylation by CDK became more important for fitness when cells were grown in chronic stress. Finally, mutants that have increased Hcm1 activity because proteasomal degradation is blocked, but retain dynamic phosphoregulation of the TAD, upregulated Hcm1 target gene expression to a greater extent and were more fit than WT cells in chronic stress. These results demonstrate that simple Hcm1 inactivation is not the mechanism by which cells survive in chronic stress; instead, dynamic regulation of Hcm1 activity – obtained through a combination of phosphorylation by CDK and dephosphorylation by CN – is critical to maintain fitness.

## Results

### Expression of a phosphomimetic Hcm1 mutant decreases fitness in LiCl stress

CDK phosphorylates eight sites in the Hcm1 TAD to activate the protein during an unperturbed cell cycle [18]. When cells are exposed to a CN-activating stressor such as $CaCl_2$ or LiCl, these phosphates are removed by the phosphatase CN [15] (Fig 1A). This dephosphorylation occurs rapidly after exposure to LiCl [15] and, notably, Hcm1 remains in a hypophosphorylated state after cells adapt to LiCl stress and resume cycling (Fig 1B). To determine if dephosphorylation of Hcm1 is necessary for cells to survive when faced with chronic LiCl stress, we utilized a constitutively active phosphomimetic Hcm1 mutant, Hcm1-8E, in which each CDK site in the TAD is mutated to two glutamic acids (S/T-P to E-E) to mimic the charge of a phosphate [18,20]. The Hcm1-8E protein is more active than WT Hcm1 and, as a result, confers a fitness advantage to cells in a competitive growth assay in optimal growth conditions [15,20]. To determine the consequence of elevated Hcm1 activity in stress, we used the same competitive growth assay to compare the fitness of cells expressing Hcm1-8E to cells expressing WT Hcm1, in the presence or absence of LiCl (Figs 1C, 1D, S1A–S1C). As previously shown [20], *hcm1-8E* cells exhibited a significant fitness benefit in the absence of stress (Fig 1C and S4 Dataset). However, *hcm1-8E* cells lost their fitness advantage and displayed a modest, but not statistically significant, fitness defect when cultured in medium containing LiCl (Fig 1D and S4 Dataset), supporting the possibility that cells need to inactivate Hcm1 to maximize fitness in the presence of chronic LiCl stress.

### Hcm1 mutants with fixed phosphorylation states are less fit in LiCl stress

Hcm1 retains some phosphorylation when cells are grown for three days in LiCl (Fig 1B), suggesting that some activity may remain, and that a reduced level of Hcm1 activity might be optimal in stress. To test this hypothesis, we employed

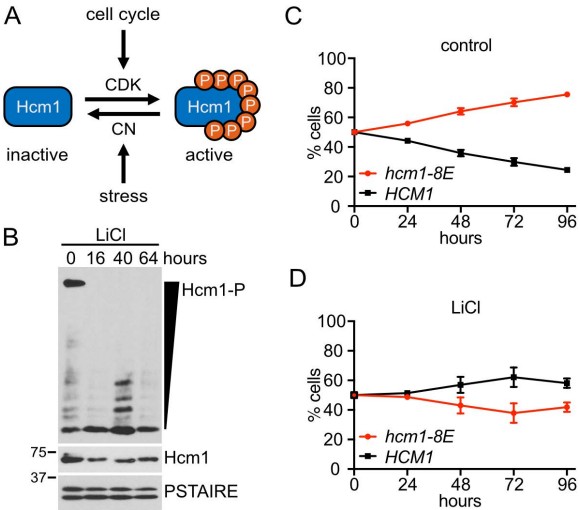

**Fig 1. Expression of a phosphomimetic Hcm1 mutant decreases fitness in LiCl stress. (A)** Hcm1 activity is regulated by cyclin-dependent kinase (CDK) and the phosphatase calcineurin (CN). **(B)** Phos-tag and standard Western blots showing Hcm1 phosphorylation and expression after the indicated number of hours in LiCl stress. Hcm1 was detected with an antibody that recognizes a 3V5 tag, PSTAIRE is shown as a loading control. Representative blots from n = 3 experiments are shown. **(C-D)** Strains with the indicated genotypes were co-cultured in control media (C) or media with 150mM LiCl (D). Percentage of each strain was quantified by flow cytometry at the indicated timepoints. An average of n = 3 biological replicates is shown. Error bars represent standard deviations. Statistical analysis is included in S4 Dataset.

Phosphosite Scanning, a recently developed approach that can simultaneously determine the effects of hundreds of phosphosite mutations on cellular fitness [20]. Phosphosite scanning was previously used to screen a collection of Hcm1 mutants in which each CDK phosphosite (S/T-P) in the TAD is mutated to an unphosphorylatable alanine (A-P) or two glutamic acids (E-E), in all possible combinations (A/E library, Fig 2A) [20]. In unstressed growth conditions, fitness values conferred by these Hcm1 mutants are highly correlated with the activity of each mutant and this collection of mutants represents the entire continuum of possible Hcm1 activities, from the completely inactive Hcm1-8A mutant (with all phosphosites mutated to A-P) to the Hcm1-8E mutant that has increased activity relative to WT [20]. To determine the optimal amount of activity in stress, we used the same Phosphosite Scanning approach and screened the Hcm1 A/E library in control and LiCl containing media in parallel (Fig 2B).

As in the previous study, the Hcm1 A/E library was transformed into a strain where the genomic copy of Hcm1 was expressed from a galactose-inducible promoter. At the start of the screen, cells growing in galactose were diluted into control medium or medium containing LiCl and expression of WT Hcm1 was shut off by adding dextrose, ensuring that all Hcm1 protein in the cells was derived from the plasmid library (Fig 2B). Cultures were periodically sampled and diluted over the course of 72 hours, and the relative abundance of each mutant was tracked via sequencing, as described previously [20]. Unlike the previous study where cells were diluted every 12 hours to maintain logarithmic growth, cells were diluted every 24 hours. Because of this modification, cells in control conditions approached saturation and exited the cell cycle between time points (S1D Fig). However, Hcm1 expression resumed following each dilution when cells reentered the cell cycle (S1E Fig), and selection coefficients of mutants screened in control conditions using 12- and 24-hour intervals were highly correlated (S2A–S2C Fig). Thus, the modified screening protocol accurately reflects the relative fitness values of Hcm1 mutants.

We first investigated how overall fitness of each mutant, relative to cells expressing WT Hcm1, was impacted when cells were growing in stress. Selection coefficients of each mutant in LiCl-containing medium were directly compared with those from control medium (Fig 2C). Surprisingly, although there was a strong correlation between selection coefficients

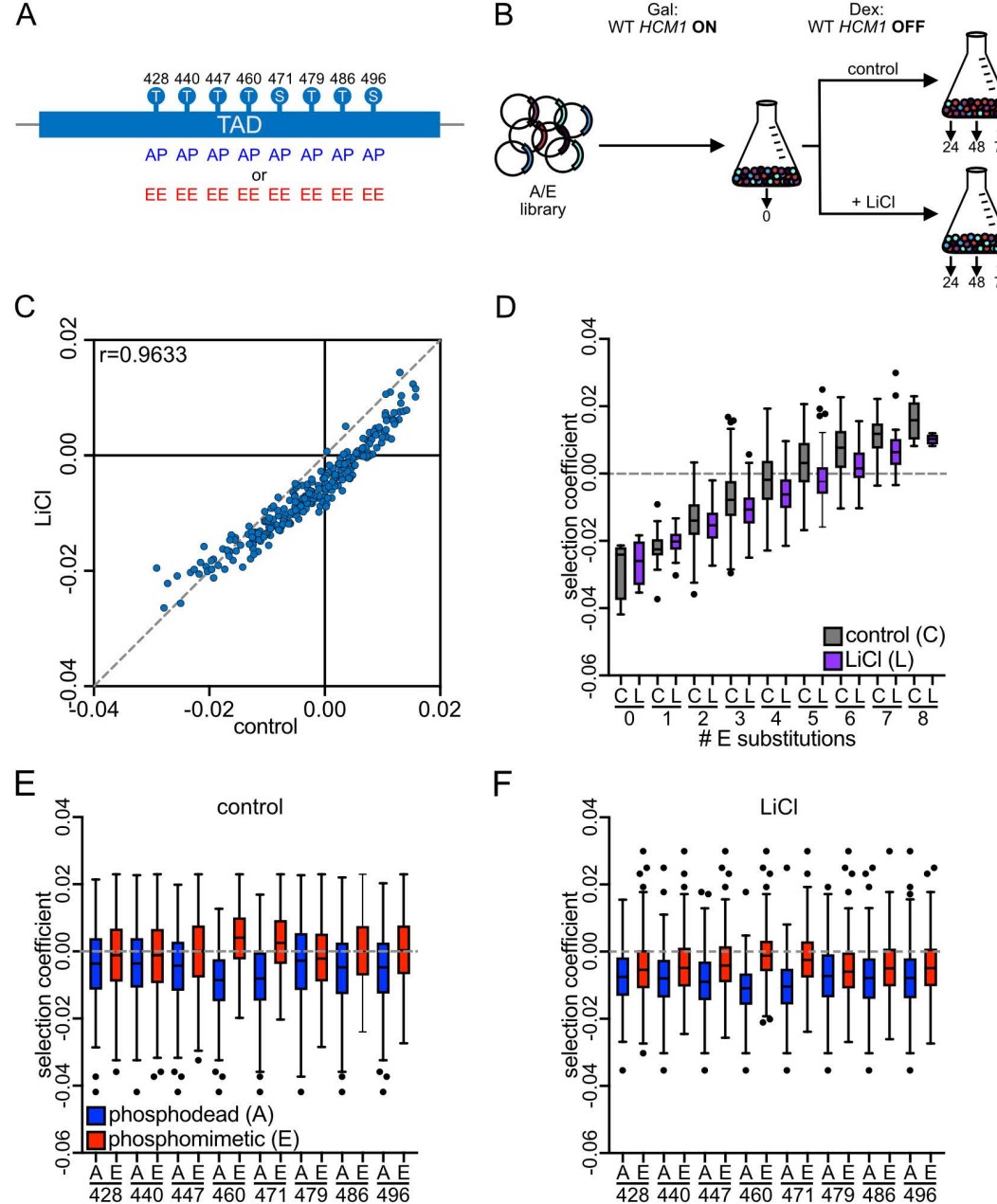

**Fig 2. Phosphosite mutations in Hcm1 decrease fitness in stress. (A)** Schematic of the Hcm1 A/E library. **(B)** Schematic of Phosphosite Scanning screens. Plasmids expressing all 256 mutants in the A/E library, as well as WT *HCM1*, were transformed into a strain in which expression of the genomic copy of *HCM1* is controlled by a galactose-inducible promoter. The pooled population growing in galactose was diluted and split into dextrose containing media (to shut-off expression of the endogenous copy of WT *HCM1*) with or without 150mM LiCl at the start of the experiment. **(C)** Scatterplot comparing average selection coefficients for each mutant in the A/E library in control and LiCl media. Pearson correlation (r) is indicated. **(D-F)** Box and whisker plots comparing the selection coefficients of different groups of mutants. The black center line indicates the median selection coefficient, boxes indicate the 25th-75th percentiles, whiskers represent 1.5 interquartile range (IQR) of the 25th and 75th percentile, black circles represent outliers. In all panels, selection coefficients are an average of n = 4 biological replicates. (D) shows selection coefficients in cells with the indicated number of phosphomimetic mutations in control or LiCl conditions. (E-F) show selection coefficients of mutants with either phosphodead or phosphomimetic mutations at each position, in control (E) or LiCl containing medium (F). Statistics for panels D-F are included in S2 Dataset.

in the two environments, almost all mutants showed a stronger reduction in fitness, relative to WT, in LiCl conditions than they exhibited in control conditions. This effect was most severe for mutants with greatest number of phosphomimetic (activating) mutations, whereas mutants that had two or fewer phosphomimetic mutations were most similar between control and LiCl media (Figs 2D and S2D). Notably, although the *hcm1-8E* mutant displayed a modest fitness defect in pairwise competition assays carried out in LiCl (Fig 1D), it had a slight fitness advantage in pooled screens (Fig 2D, mutant with eight E substitutions). This observation is consistent with previous findings that pooled screens result in higher selection coefficients than pairwise competition assays and is likely due to technical differences in the experimental approach [20]. However, the *hcm1-8E* mutant fitness advantage was modest in pooled LiCl screens and decreased in LiCl compared to control conditions. These data demonstrate that, contrary to our expectation, Hcm1 activity is required for fitness in stress. However, the most highly active mutants exhibit reductions in fitness compared to control growth conditions.

Next, we wanted to determine how individual TAD phosphosites impact cellular fitness in LiCl stress. To do this, we compared the selection coefficients of all mutants that had either a non-phosphorylatable alanine (A) or two glutamic acids (E) at each site. In both control and LiCl conditions, a phosphomimetic mutation at any site increased fitness relative to an alanine substitution at the same site (Fig 2E and 2F, compare red and blue boxes). Notably, mutations at sites T460 and S471 had the greatest effect on fitness in both conditions, consistent with previous measurements in the absence of stress [20]. Together, these data show that when phosphorylation patterns of Hcm1 are fixed because all sites are changed to either phosphodeficient or phosphomimetic amino acids, the fitness of almost all mutants decreases in stress relative to WT. This raises the possibility that it is not inactivation of Hcm1 that is important for cells to maintain fitness in stress, but rather dynamic regulation conferred by phosphorylation and dephosphorylation of the TAD.

## Requirement for Cks1-dependent priming is elevated in stress

If dynamic Hcm1 regulation is critical for stress survival, we hypothesized that mechanisms enhancing CDK phosphorylation would be especially important under stress. One such mechanism is kinase priming by the CDK accessory subunit Cks1, which binds phosphorylated threonines near the N-terminus of a multisite phosphorylated domain and promotes phosphorylation of downstream CDK sites [21,22]. Phosphosite Scanning can reveal Cks1-dependent regulation by assessing how mutations at one site—either non-phosphorylatable alanine or phosphomimetic glutamic acid—affect phosphorylation at others [20]. To test this, we screened two additional Hcm1 mutant libraries: one with WT or alanine substitutions (WT/A, Fig 3A), and one with WT or phosphomimetic substitutions (WT/E, Fig 4A). We compared the fitness of all mutants relative to WT in both control and LiCl stress conditions.

We first performed Phosphosite Scanning of the WT/A library under control and LiCl stress conditions. Since Cks1 binds only phosphothreonine [21,22], alanine substitutions impair activity both by preventing phosphorylation and by blocking Cks1 binding, which reduces phosphorylation of downstream sites. As in the A/E screen, selection coefficients of WT/A mutants were correlated between conditions, but all scores were lower in LiCl (Fig 3B). Fitness was more strongly reduced by alanine substitutions in a WT background (WT/A) than in a phosphomimetic background (A/E), consistent with Cks1-dependent regulation [20]. Despite the overall fitness reduction in the WT/A library, fitness increased with the number of WT sites (Figs 3C and S2E), and WT residues were consistently more beneficial than alanine at each position (Fig 3D and 3E), in both conditions. These results support the conclusion that phosphorylation at each TAD site contributes to Hcm1 activity in both control and stress environments.

To further assess the role of Cks1 priming in stress, we screened the WT/E library (Fig 4A). Like alanine mutations, phosphomimetic (EE) substitutions cannot serve as Cks1 priming sites. While they retain charge and partially activate Hcm1, they block Cks1 binding. Thus, TP-to-EE mutations at a priming site can reduce fitness by disrupting downstream phosphorylation at WT sites. Additionally, if priming is of increased importance in stress, this reduction in fitness seen with EE substitutions in the WT/E library might be enhanced. This is indeed what was observed: most mutants were more fit than WT in control conditions but showed reduced fitness under LiCl stress (Fig 4B, lower right quadrant). However,

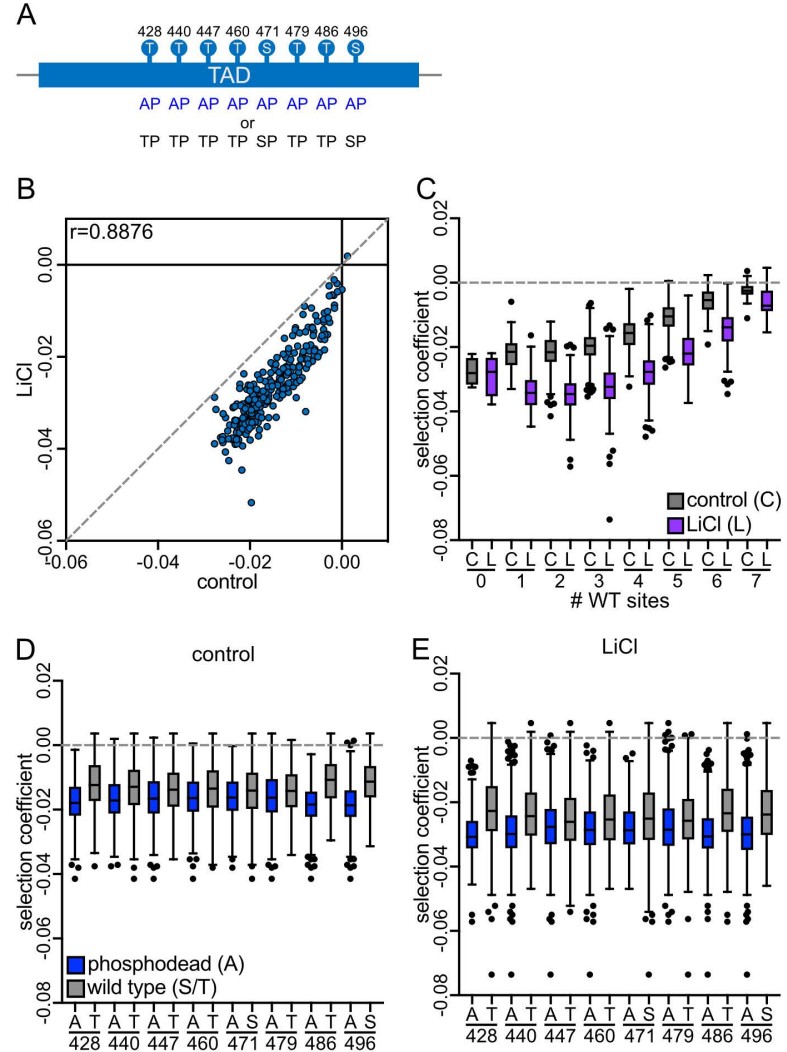

**Fig 3. All phosphosites in the Hcm1 TAD contribute to fitness in stress. (A)** Schematic of the Hcm1 WT/A library. **(B)** Scatterplot comparing average selection coefficients for each mutant in the WT/A library in control and LiCl media. Pearson correlation (r) is indicated. **(C-E)** Box and whisker plots comparing the selection coefficients of different groups of mutants. The black center line indicates the median selection coefficient, boxes indicate the 25th-75th percentiles, whiskers represent 1.5 interquartile range (IQR) of the 25th and 75th percentile, black circles represent outliers. In all panels, selection coefficients are an average of n = 4 biological replicates. (C) shows selection coefficients in cells with the indicated number of WT sites in control or LiCl conditions. (D-E) show selection coefficients of mutants that are either WT (S or T) or alanine at each position, in control (D) or LiCl containing medium (E). Statistics for panels C-E are included in S2 Dataset.

mutants containing T460E and S471E (Fig 4B, red circles) maintained higher fitness in stress, likely because these key sites do not depend on upstream priming. Furthermore, fitness in LiCl did not correlate with the number of phosphomimetic mutations (Figs 4C and S2F), suggesting that Cks1-mediated priming plays a critical role during stress.

To pinpoint Cks1 priming sites that are important during stress, we analyzed the fitness of phosphomutants based on their genotype at each site. Under non-stress conditions, a phosphomimetic mutation that reduces fitness—reflected by a lower selection coefficient compared to the WT CDK site—suggests that the site may function as a Cks1 priming site [20]. For example, in control conditions a T428E mutation slightly reduces fitness in the WT/E library (Fig 4D) but increases fitness relative to T428A in the A/E library (Fig 2E), supporting its role as a Cks1 priming site [20]. The negative fitness effect

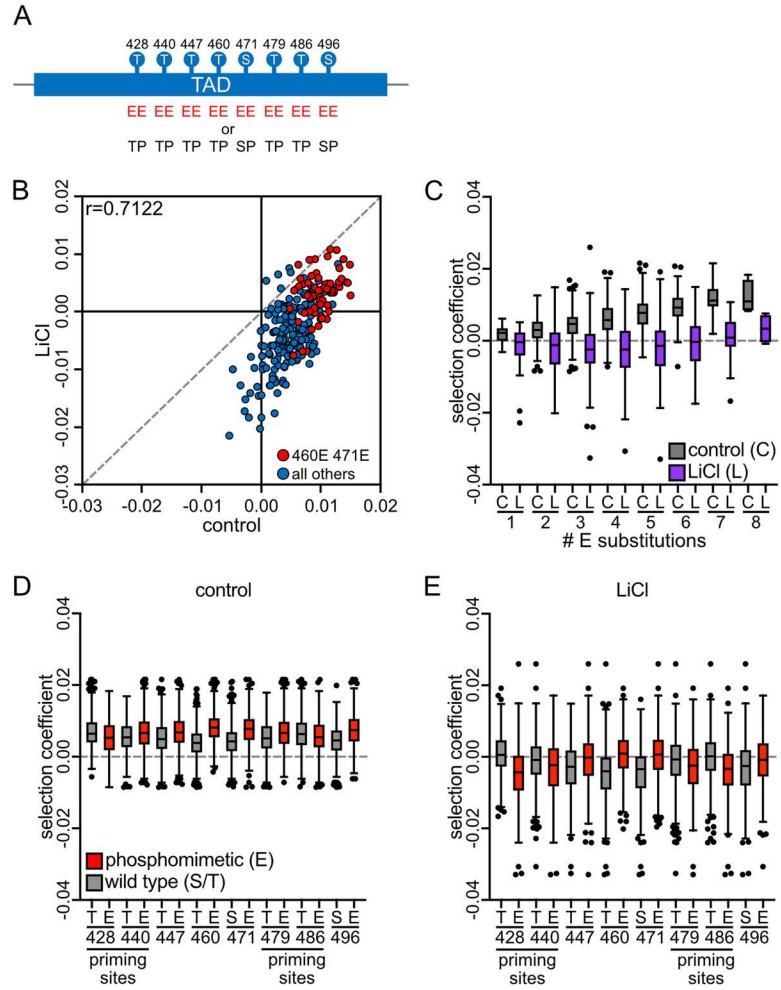

**Fig 4. Elevated importance of Cks1 priming in stress. (A)** Schematic of the Hcm1 WT/E library. **(B)** Scatterplot comparing average selection coefficients for each mutant in the W/E library in control and LiCl media. Pearson correlation (r) is indicated. Red represents mutants that are phospho-mimetic at sites T460 and S471, blue represents all other mutants. **(C-E)** Box and whisker plots comparing the selection coefficients of different groups of mutants. The black center line indicates the median selection coefficient, boxes indicate the 25th-75th percentiles, whiskers represent 1.5 interquartile range (IQR) of the 25th and 75th percentile, black circles represent outliers. In all panels, selection coefficients are an average of n = 4 biological repli-cates. (C) shows selection coefficients in cells with the indicated number of phosphomimetic mutations in control or LiCl conditions. (D-E) show selection coefficients of mutants that are either WT (S or T) or phosphomimetic at each position, in control (D) or LiCl containing medium (E).

of T428E is more pronounced under LiCl stress, and a similar reduction is observed with the T440E mutation (Fig 4E). Interestingly, T479E and T486E mutations also reduce fitness in LiCl, despite having neutral or slightly positive effects in control conditions (Fig 4D). These findings suggest that T479 and T486 may also act as Cks1 priming sites within the Hcm1 TAD, with their importance becoming evident only under stress conditions.

To investigate this effect further, we assessed the fitness of all mutants containing T428E T440E or T479E T468E in the WT/E screen. These mutants showed significantly reduced fitness under stress conditions compared to those with WT sites at these positions, despite similar fitness in control conditions (Fig 5A). Additionally, a previously characterized mutant with disruption of the N-terminal Cks1 priming sites (T428S T440S T447S, *hcm1-3S*) exhib-ited reduced fitness in both conditions (Fig 5B–5D), consistent with the observation that mutations detrimental in

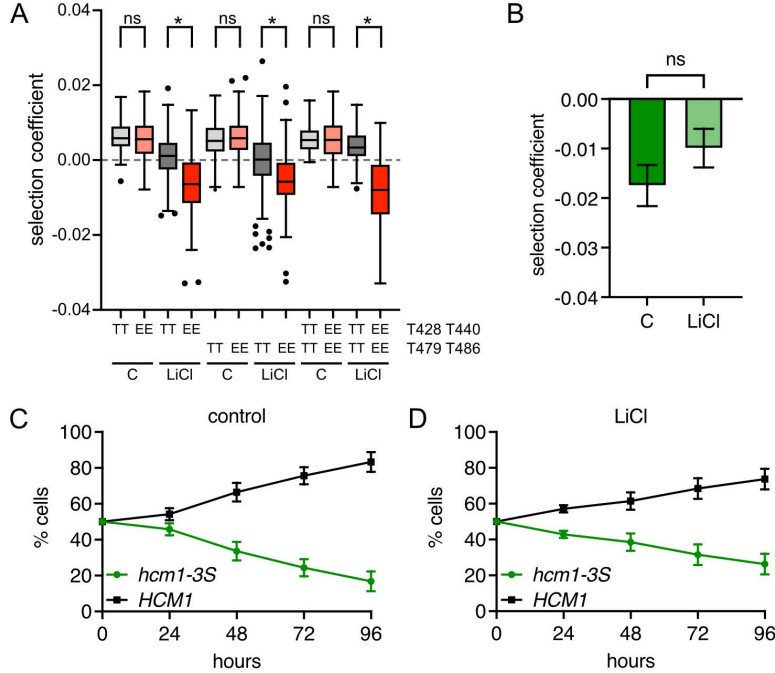

**Fig 5. Cks1 priming sites promote fitness in stress. (A)** Box and whisker plots comparing the selection coefficients of mutants that are WT (TT) or phosphomimetic (EE) at indicated positions. The black center line indicates the median selection coefficient, boxes indicate the 25th-75th percentiles, whiskers represent 1.5 interquartile range (IQR) of the 25th and 75th percentile, black circles represent outliers. In all panels, selection coefficients are an average of n = 4 biological replicates. **(B)** Comparison of selection coefficients of the Cks1-priming site mutant, *hcm1-3S*, in control and LiCl conditions. A paired t-test was used to determine that selection coefficients are not significantly different (ns) between conditions. **(C-D)** *HCM1* and *hcm1-3S* strains were co-cultured in control media (C) or media with 150mM LiCl (D). Percentage of each strain was quantified by flow cytometry at the indicated time-points. An average of n = 6 biological replicates is shown. Error bars represent standard deviations.

non-stress conditions also impair fitness under stress (Fig 2D). Together, these results support the conclusion that dynamic phosphorylation of the Hcm1 TAD—facilitated by Cks1 priming—is essential for maintaining fitness during LiCl stress.

## Dynamic phosphorylation of the Hcm1 TAD is required for fitness in stress

If the *hcm1-8E* mutant is less fit in stress because its phosphorylation cannot be dynamically regulated, and not because it has increased activity, then an alternative mutant in which Hcm1 activity is increased by a different mechanism might not exhibit reduced fitness in stress. To test this possibility, we examined the fitness of *hcm1-3N* mutant cells. Hcm1-3N contains three alanine substitutions in the N-terminal phosphodegron that prevent proteasomal degradation and stabilize the protein [18], thereby increasing Hcm1 activity without perturbing TAD phosphorylation dynamics (Fig 6A). First, we compared the activity of the Hcm1 mutants directly, by expressing each mutant in an *hcm1Δ* reporter strain, where two copies of the Hcm1-responsive element from the *WHI5* promoter drive GFP expression. Notably, Hcm1-3N was more active than WT Hcm1, although it was less active than Hcm1-8E (Fig 6B). Despite this difference, both *hcm1-3N* and *hcm1-8E* cells exhibited similar increases in fitness in the absence of stress (Figs 1C, 6C and 6D) [15,20]. If dynamic regulation of the Hcm1 TAD is important for fitness in stress, *hcm1-3N* cells should differ from *hcm1-8E* and retain their fitness advantage.

To test this hypothesis, pairwise competition assays were carried out between *hcm1-3N* and WT cells in control and LiCl containing medium. Notably, in contrast to the decreased fitness observed in *hcm1-8E* cells challenged with LiCl stress (Figs 1D and 6C), the fitness benefit in *hcm1-3N* cells was enhanced under stress conditions (Fig 6C and 6E).

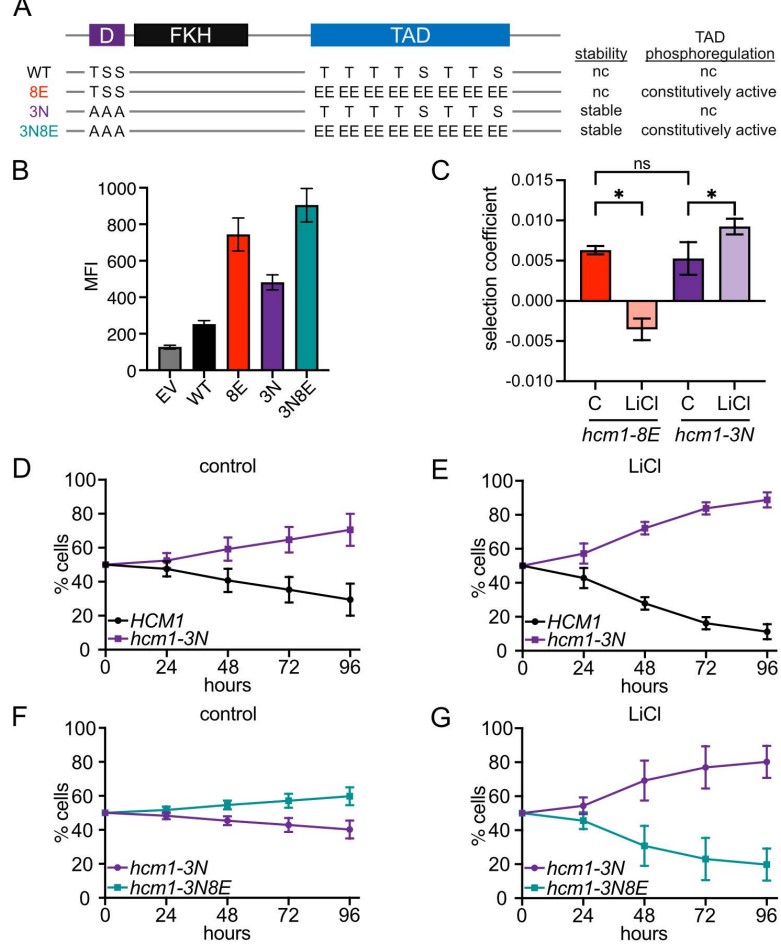

**Fig 6. Dynamic phosphorylation of the Hcm1 TAD promotes fitness in stress. (A)** Diagram of Hcm1 mutant proteins showing mutated phospho-sites and impacts on protein stability and phosphoregulation of the TAD region. nc, no change. **(B)** Mean Fluorescence Intensity (MFI) of GFP expression in Hcm1 reporter strains expressing the indicated Hcm1 proteins or an empty vector (EV). An average of n = 6 replicates is shown. Error bars represent standard deviation. **(C)** Comparison of average selection coefficients of the indicated strains and growth conditions, from pairwise assays shown in Figs 1C, 1D, 6D, and 6E. One-way ordinary ANOVA with Šídák's multiple comparisons test was used to test significance, *p < 0.0001. **(D-E)** HCM1 and hcm1-3N strains were co-cultured in control media (D) or media with 150mM LiCl (E). Percentage of each strain was quantified by flow cytometry at the indicated timepoints. An average of n = 13 biological replicates is shown. Error bars represent standard deviations. **(F-G)** hcm1-3N and hcm1-3N8E strains were co-cultured in control media (F) or media with 150mM LiCl (G). Percentage of each strain was quantified by flow cytometry at the indicated timepoints. An average of n = 13 biological replicates is shown. Error bars represent standard deviations.

To compare these effects directly, these two sets of mutations were combined to generate a stable, constitutively active mutant (*hcm1-3N8E*, Fig 6A and 6B). In a pairwise competition assay, *hcm1-3N8E* cells had similar fitness as *hcm1-3N* cells in control conditions (Fig 6F), as previously reported [20]. However, *hcm1-3N8E* cells were less fit than *hcm1-3N* in LiCl stress (Fig 6G). Therefore, preventing dynamic phosphorylation of the Hcm1-3N protein reverses the fitness benefit provided by its stabilization and increased expression in stress.

## Dynamic phosphorylation enhances Hcm1 activity in stress

When cells are continuously exposed to a CN-activating stress, cells experience bursts in cytosolic $Ca^{2+}$, followed by pulses of CN activity [23,24]. This suggests that Hcm1 may undergo pulses of inactivation in chronic stress, which are

then reversed by CDK activity. For some TFs, increasing the frequency of pulses of activation results in a greater induction of target gene expression, compared to increasing the amplitude of TF activity [25,26]. Therefore, we considered that Hcm1 may undergo pulses of activity in stress, through modulation of phosphorylation, which in turn could further increase the expression of its target genes and promote fitness.

To support this model, we first examined the dynamics of CN activation in response to LiCl using a fluorescent CN reporter [10]. CN activity peaked within five minutes of LiCl addition and then rapidly declined (S3A–S3C Fig). However, sporadic activation persisted throughout the 72-hour time course (S3B Fig). Notably, after the initial activation wave subsided, brief pulses of CN activity—lasting five minutes or less—were observed in individual cells (S3D Fig). These findings suggest that Hcm1 may undergo intermittent dephosphorylation and inactivation during LiCl stress.

If pulses of Hcm1 dephosphorylation and re-phosphorylation stimulate its activity in stress, the prediction is that WT and *hcm1-3N* cells should display an increase in Hcm1 target gene expression, compared to unstressed conditions, whereas *hcm1-8E* and *hcm1-3N8E* should not. To test this possibility, we evaluated the activity of each mutant in the Hcm1 reporter strain described above, in control and LiCl conditions. Strains were passaged in monoculture, using the same dilution protocol that was employed for fitness assays, and GFP levels were measured using both flow cytometry and Western blotting after 40 hours of growth. As predicted, Hcm1-3N exhibited a significant increase in activity under stress, whereas Hcm1-8E and Hcm1-3N8E showed either decreased or unchanged activity (Figs 7A, S4A and S4B). The WT protein also appeared more active under stress based on GFP levels detected by Western blot (S4B Fig). However, flow cytometry revealed a modest fluorescence increase in both WT and EV cells (Figs 7A and S4A). Since no GFP protein was detected by Western blot in EV cells, we conclude that the fluorescence increase may be due to a slight Hcm1-independent increase in autofluorescence under stress. Notably, the increased activity of WT and Hcm1-3N could not be attributed to

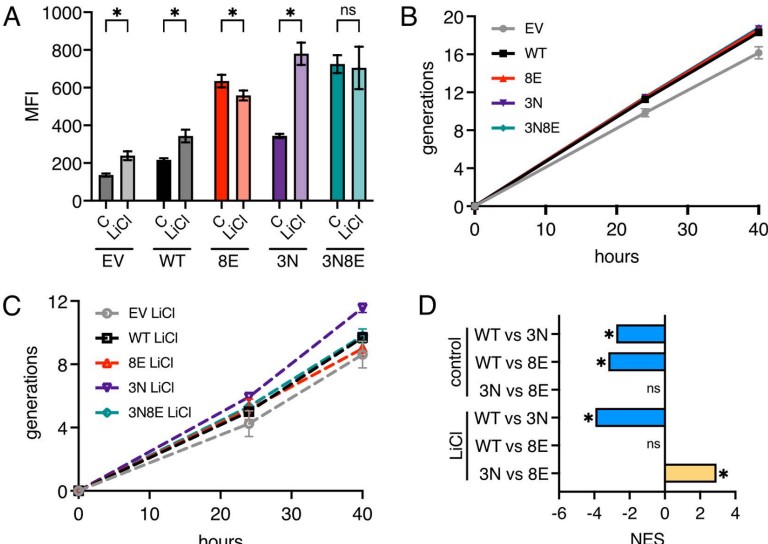

**Fig 7. Dynamic phosphorylation increases Hcm1 activity in stress. (A)** Mean Fluorescence Intensity (MFI) of GFP expression in Hcm1 reporter strains expressing the indicated Hcm1 proteins or an empty vector (EV), after 40 hours of growth in control medium or medium with 150mM LiCl. An average of n = 6 replicates is shown. Error bars represent standard deviation. Two-way ANOVA with Šídák's multiple comparisons test was used to test significance, *p < 0.05. Representative histograms are shown in S4A Fig. **(B-C)** Cumulative generations of strains from (A) over the 40-hour experiments. Shown is an average of n = 6 replicates, errors bars represent standard deviations. Strains grown in control conditions are shown in part (B). Strains grown in LiCl stress are shown in part (C). **(D)** Normalized enrichment scores (NES) of Hcm1 target genes from GSEA analysis of the indicated comparisons. Asterisk (*) indicates FDR = 0, ns indicates not significant.

changes in Hcm1 protein levels (S4B Fig) or differences in the proportion of cells in S-phase (S4C Fig). These data support the model that pulses of activation increase Hcm1 target gene expression.

We also examined whether Hcm1 mutants could enhance proliferation by comparing population growth when the different Hcm1 proteins were expressed in the *hcm1Δ* reporter strain during the 40-hour time course. In the absence of stress, all mutants supported proliferation as well as WT Hcm1, whereas empty vector (EV) cells that do not express Hcm1 exhibited a proliferation defect (Fig 7B). In contrast, mutants differentially affected proliferation when grown in stress. Although all strains underwent fewer doublings in stress, strains expressing WT, Hcm1-8E and Hcm1-3N8E were more similar to EV cells, whereas Hcm1-3N cells displayed a proliferative advantage (Fig 7C). These results support the conclusions from pairwise competition assays (Figs 1C, 1D, and 6C–6G) and show that Hcm1-3N expression provides a fitness advantage in stress.

We next examined how Hcm1 mutants impact the expression of endogenous Hcm1 target genes in chronically stressed cells. Hcm1 only activates target gene expression during S-phase, so the clearest way to examine the effect of Hcm1 mutants is to compare target gene expression in S-phase synchronized cells [18]. However, we found that after cells have adapted to growing in LiCl, they do not release synchronously from a G1 arrest (S5A Fig), making it difficult to quantify gene expression specifically in S-phase. Therefore, we quantified target gene expression by sequencing mRNA from asynchronous cells that were grown in control medium or LiCl for 40 hours. Because the experiment was performed in asynchronous cultures, where only ~16% of cells are in S-phase (S5B Fig), we observed modest fold change differences between wild type and mutant strains for individual Hcm1 target genes (S3 Dataset). For this reason, we used Gene Set Enrichment Analysis (GSEA) to examine changes in expression of Hcm1 target genes as a group. GSEA revealed that Hcm1 targets were collectively expressed at elevated levels in both *hcm1-3N* and *hcm1-8E* mutants compared to WT in control conditions, confirming that the two mutants exhibit a similar increase in activity (Fig 7D, a negative normalized enrichment score (NES) indicates lower expression in WT compared to mutant). However, when cells were grown in LiCl, Hcm1 target genes increased in expression in *hcm1-3N* cells, but not in *hcm1-8E* cells. Moreover, target genes were more highly expressed in *hcm1-3N* cells than *hcm1-8E* cells when they were directly compared (Fig 7D, a positive NES indicates higher expression in *hcm1-3N* compared to *hcm1-8E*). These results demonstrate that dynamic phosphorylation of Hcm1 in stress increases expression of Hcm1 target genes and suggests that expression of these genes promotes fitness in stress.

## Discussion

Immediately upon exposure to an environmental stressor, cells rewire many cellular pathways to promote stress resistance and long-term survival. A conserved feature of this acute response is the inactivation of cell cycle-regulatory TFs and downregulation of their target genes. Although cell cycle arrest is not required for the execution of the acute stress response [27], it is possible that arrest and/or downregulation of cell cycle-regulatory genes is important for adaptation and survival after cells resume proliferation in the new environment. In support of this possibility, we found that expression of the Hcm1-8E phosphomimetic mutant, which cannot be dephosphorylated and inactivated, reduces fitness in chronic LiCl stress (Fig 1D). However, several pieces of evidence argue that simple Hcm1 inactivation does not promote fitness in stress, but rather its activity must be toggled on and off to maximally activate target gene expression and ensure fitness. First, almost all Hcm1 phosphosite mutants that retain activity but have fixed phosphorylation states are less fit, relative to WT, in LiCl than control conditions (Fig 2C). This includes mutants that have WT-like activity, as well as those that have increased activity compared to WT. Second, TP motifs that function as Cks1 priming sites to promote phosphorylation by CDK are of greater importance in stress (Fig 5A), which suggests that increased CDK-dependent phosphorylation is required to counteract CN-dependent dephosphorylation. This is seen for T428 and T440, which are previously characterized Cks1 priming sites [20], as well as two putative Cks1 priming sites T479 and T486, which are of increased importance only in stress. Finally, cells expressing the Hcm1-3N protein, which has increased activity because it is stabilized

but retains phosphorylation-dependent activation [20], promotes the highest level of target gene expression and confers a fitness advantage in LiCl stress (Figs 6E and 7D). Together, these data demonstrate that Hcm1 activity is required for fitness in stress, and that the ability to add and remove phosphates is critical for maximal activity.

Dynamic regulation by phosphorylation is a recognized feature of many TFs, most notably TFs that respond to stress. In mammalian cells, pulsatile nuclear localization and activation of NFAT [28], p53 [29–31] and NFκB [32–35] lead to an altered transcriptional output in response to different signals. In budding yeast, at least ten TFs display pulsatile nuclear localization in response to specific cues, thereby increasing frequency of their activation [36]. In the case of the stress-activated TF Msn2, exposure to distinct stressors triggers either sustained or pulsatile Msn2 nuclear localization [37], resulting in expression of distinct groups of target genes belonging to different promoter classes [25,26]. Notably, the CN-regulated TF Crz1 also exhibits pulsative activation via regulation of its nuclear localization [23,24]. When cells are exposed to continuous extracellular $CaCl_2$, the frequency of cytosolic $Ca^{2+}$ pulses increases, and these are followed by pulses of CN activation. CN then dephosphorylates Crz1, leading to pulses of nuclear localization and target gene activation. Importantly, Hcm1 does not display pulses of nuclear localization [36], instead we propose that its activation is controlled by pulses of phosphorylation. Since Hcm1 is also a CN target, when cells are growing in LiCl stress pulses of CN activity are expected to dephosphorylate and inactive Hcm1, which is then countered by CDK-dependent phosphorylation, leading to restoration of Hcm1 activity. To our knowledge the only described mechanism of TF frequency modulation is through changing TF localization, therefore dynamic phosphorylation of a TAD represents a novel mechanism of controlling TF target gene expression in stress.

Dynamic phosphoregulation is likely to also control the activities of other CDK target proteins in stress. In addition to CN, which antagonizes CDK as well as other kinases [13], the cell cycle-regulatory phosphatase Cdc14 is activated during the stress response [38–40] and could regulate phosphorylation dynamics. Moreover, a recent phosphoproteomic study monitored proteome-wide phosphorylation after acute exposure to more than 100 stress conditions and found that ~20% of phosphorylated proteins, including Hcm1, show a change in phosphorylation after acute stress exposure [41]. Interestingly, the two Hcm1 phosphosites that changed most frequently in that study were T479 and T486, which our data suggests may be Cks1-dependent priming sites that are of increased importance in stress (Fig 5A). Notably, monitoring rapidly changing phosphorylation patterns on individual proteins is not technically feasible, therefore observations of pulsatile phosphorylation have largely been restricted to proteins where changes in phosphorylation are measured indirectly by a change in localization [36]. Here, we show that Phosphosite Scanning can be used to infer the importance of phosphorylation dynamics [20]. Screening mutants in which all sites are mutated to either non-phosphorylatable or phospho-mimetic mutations reveals the importance of being able to add and remove phosphates. In addition, by screening mutants that combine phosphosite mutations and wild type sites, Phosphosite Scanning can reveal whether priming sites for the CDK accessory subunit Cks1 are important in stress, supporting the importance of dynamic phosphorylation. We anticipate that this approach will enable the investigation of phosphorylation dynamics of other proteins and reveal whether dynamic phosphorylation is a common mechanism that modulates protein function when cells are growing in stressful environments.

## Materials and Methods

### Yeast strains and plasmids

All cultures were grown in rich medium (YM-1) or synthetic media lacking uracil (C-Ura) with 2% dextrose or galactose. Cultures were grown at 30°C or 23°C, as indicated. A record of strains and plasmids used in this study can be found in S1 and S2 Tables, respectively. To construct the Hcm1 reporter strain, a synthetic DNA containing two copies of the previously characterized Hcm1 responsive element from the *WHI5* promoter [16] was cloned upstream of the minimal *GAL1* promoter and GFP. This sequence was integrated into a silent region on chromosome VI [42], and then introduced into an *hcm1Δ* background by genetic cross.

## Co-culture competition assays

Pairwise competition assays were done as described in Conti et al. 2023 ([20] and S1A Fig). Strains in which the endogenous copy of *HCM1* is regulated by a galactose inducible promoter (*GAL1p-HCM1*) and expressing either WT or non-fluorescent GFP (GFP-Y66F) were transformed with low-copy plasmids expressing WT or mutant *HCM1* from the *HCM1* promoter. Initially, cultures were grown in synthetic media lacking uracil with 2% galactose to ensure expression of *HCM1*. Logarithmic phase cells were equally mixed by adding one optical density (OD600) of each strain to the same culture tube in a final volume of 10mL C-Ura with 2% galactose. To determine the starting abundance of each strain, 0.15 optical densities were collected from the co-culture tubes. Samples were pelleted by centrifugation, resuspended in 2mL sodium citrate buffer (50mM sodium citrate, 0.02% $NaN_3$, pH 7.4) and stored at 4°C pending analysis by flow cytometry. To evaluate fitness in LiCl stress, co-cultures were diluted within a range of 0.005-0.04 optical densities (OD600) into synthetic media lacking uracil with 2% dextrose with or without 150mM LiCl after mixing at the start of the experiment. Cultures were then sampled and diluted every 24 hours for a total of 96 hours. Cultures reached saturation prior to dilution. At each timepoint, 0.15 optical densities were collected, pelleted, and resuspended in 2mL sodium citrate, and stored at 4°C until the conclusion of the experiment. Following the final timepoint, the percentage of GFP positive cells was quantified in each sample using a Guava EasyCyte HT flow cytometer and GuavaSoft software. 5000 cells were measured in all samples. Results were analyzed using FlowJo software. Averages of n = 3–13 biological replicates are shown; exact number is indicated in the figure legends. Selection coefficients for pairwise assays were calculated by calculating the slope of the best fit line of log2 fold change in mutant fraction over time, relative to WT in the same experiment. Control competitions were performed with WT (*HCM1*) cells expressing GFP or the non-fluorescent mutant GFP (GFP-Y66F). Expression of these markers had no effect on fitness (S1B and S1C Fig).

## Western blotting

Yeast culture amounting to one optical density (OD600) was collected, pelleted by centrifugation, and stored at -80°C prior to lysis. Cell pellets were lysed by incubation with cold TCA buffer (10mM Tris pH 8.0, 10% trichloroacetic acid, 25mM ammonium acetate, 1mM EDTA) on ice for 10 minutes. Lysates were mixed by vortexing and pelleted by centrifugation at 16,000xg for 10 minutes at 4°C. The supernatant was aspirated, and cell pellets were resuspended in 75µL resuspension solution (100mM Tris pH 11, 3% SDS). Lysates were incubated at 95°C for five minutes then allowed to cool to room temperature for five minutes. Lysates were clarified by centrifugation at 16,000xg for 30 seconds at room temperature. Supernatants were then collected, transferred to a new tube and 25µL 4X SDS-PAGE sample buffer (250mM Tris pH 6.8, 8% SDS, 40% glycerol, 20% β-mercaptoethanol) was added. The samples were incubated at 95°C for five minutes, then allowed to cool to room temperature and stored at -80°C.

For standard Western blots, resolving gels contain 10% acrylamide/bis solution 37.5:1, 0.375M Tris pH 8.8, 0.1% SDS, 0.1% ammonium persulfate (APS), 0.04% tetremethylethylenediamine (TEMED). Phos-tag gels contain 6% acrylamide/bis solution 29:1, 386mM Tris pH 8.8, 0.1% SDS, 0.2% APS, 25µM Phos-tag acrylamide (Wako), 50µM manganese chloride and 0.17% TEMED. All stacking gels contain 5% acrylamide/bis solution 37.5:1, 126mM Tris pH 6.8, 0.1% SDS, 0.1% APS and 0.1% TEMED. All SDS-PAGE gels were run in 1X running buffer (200mM glycine, 25mM Tris, 35mM SDS). Phos-tag gels were washed twice with 1X transfer buffer containing 10mM EDTA for 15 minutes (150mM glycine, 20mM Tris, 1.25mM SDS, 20% methanol) and once with 1X transfer buffer for 10 minutes on a shaking platform with gentle agitation. All gels were transferred to nitrocellulose in cold 1X transfer buffer at 0.45A for two hours. After transfer, nitrocellulose membranes were blocked in a 4% milk solution for 30 minutes.

Western blotting was performed with primary antibodies that recognize a V5 epitope tag (Invitrogen, 1:1000 dilution), PSTAIRE (P7962, Sigma, 1:10,000 dilution), G6PDH (A9521, Sigma, 1:10,000 dilution), and GFP (632381, Clontech, 1:1000). Primary antibody incubations were performed overnight at 4°C. Importantly, molecular weight makers are not shown with Phos-tag gels as they do not accurately reflect the molecular weight of proteins.

## Phosphosite scanning screens

Phosphosite scanning screens were carried out using pooled plasmid libraries that were previously constructed and characterized [20], and transformed into a *GAL1p-HCM1* strain. A plasmid expressing WT *HCM1* was added to all libraries for normalization. During transformation, cells were cultured in YM-1 containing 2% galactose to maintain expression of endogenous *HCM1.* Following transformation, cells were cultured overnight at 23°C in synthetic media lacking uracil (C-Ura) with 2% galactose. After approximately 16 hours, an aliquot of transformed cells was removed and plated on C-Ura to confirm a transformation efficiency of at least 10X library size. Remaining cells were washed with 15mL C-Ura with 2% galactose five times, resuspended in 50mL C-Ura with 2% galactose and allowed to grow to logarithmic phase for approximately 48 hours at 30°C. The starting population was sampled to determine the initial abundance of each mutant in the population prior to selection. Cell pellets amounting to 20 optical densities were harvested, frozen on dry ice, and stored at -80°C prior to preparation of sequencing libraries. To evaluate fitness in stress, cultures were then diluted into synthetic media lacking uracil with 2% dextrose with or without 150mM LiCl after sampling at time zero. For all timepoints after time zero, cells were diluted into a range of 0.08 and 0.1 optical densities in 10mL of the appropriate media. Cultures were sampled and diluted as above every 24 hours for a total of 72 hours.

## Illumina sequencing library preps

For analysis by sequencing, plasmids were recovered from the frozen samples using a YeaStar Genomic DNA Kit (Zymo Research). Mutant *hcm1* sequence was amplified by PCR (21 cycles) using plasmid specific primers and Phusion High-Fidelity DNA polymerase (New England Biolabs). DNA fragments were purified from a 1% agarose gel using a QIAquick Gel Extraction Kit (Qiagen). Barcoded TruSeq adapters were added to the mutant fragments by PCR (7 cycles) using primers specific to the *HCM1* region fused to either the TruSeq universal adapter or to a unique TruSeq indexed adapter. Sequences of oligonucleotides that were used in library construction can be found in S3 Table. Barcoded fragments were purified from a 1% agarose gel as described above. Pooled barcoded libraries were sequenced on a HiSeq4000 platform (Novogene) to obtain paired-end 150 base pair sequencing reads. All sequencing data is available from the NCBI Sequencing Read Archive under BioProject # PRJNA1117860.

## Phosphosite scanning data analysis

Abundance of *HCM1* alleles was quantified by counting all paired-end sequencing fragments that had an exact match to an expected sequence in both reads using a custom python script. Custom scripts used to generate count tables are available on GitHub (https://github.com/radio1988/mutcount2024/tree/main/AE_type) and Zenodo (https://zenodo.org/records/13144766). Selection coefficients were calculated as the slope of the log2 fraction of reads versus time for each mutant, normalized to the log2 fraction of reads versus time of WT. All selection coefficients for all screens can be found in S1 Dataset. Box and whisker plots were generated using GraphPad Prism software. In all box and whisker plots the black center line indicates the median selection coefficient, boxes indicate the 25th-75th percentiles, black lines represent 1.5 interquartile range (IQR) of the 25th and 75th percentile, black circles represent outliers. Statistical analyses for all plots of Phosphosite Scanning data are included in S2 Dataset.

## CN activation in LiCl stress

To monitor calcineurin (CN) activation, we used a previously characterized fluorescent reporter (CNR-C), which is a fragment of Crz1 that lacks the DNA-binding domain fused to GFP [10]. CNR-C translocates to the nucleus upon CN activation. As a control, CNR-C was expressed in a strain lacking the CN regulatory subunit (*cnb1Δ*). Cells were diluted at 8- and 16-hour timepoints to maintain logarithmic growth, in synthetic complete media supplemented with 2% dextrose and 0.15M LiCl. Dilutions were made to an OD600 of 0.08 and 0.008, respectively. At each timepoint, an OD600 of 0.1 was collected, centrifuged, and resuspended in a small volume of SC media for imaging.

Microscopy was performed using a Zeiss AxioObserver 7 microscope equipped with a Hamamatsu Orca Fusion-BT camera. Bright-field images were acquired at 30% intensity with a 40ms exposure, while GFP fluorescence images were captured at 30% intensity, with a 300ms exposure. All images were processed with identical exposure settings and uniform adjustments to brightness and contrast. Nuclear localization of CNR-C in S3B Fig was quantified by manual inspection using ImageJ. For each sample and timepoint, a minimum of 100 cells were scored for nuclear GFP signal.

## Time-lapse imaging of CN activity

Time-lapse imaging of CNR-C was performed using a Zeiss AxioObserver 7 microscope equipped with a Hamamatsu Orca Fusion-BT camera and a temperature-controlled chamber. Log-phase cells (OD600 = 0.4–0.8) were collected, centrifuged, and resuspended in synthetic complete media. Cells were then plated onto a Concanavalin A (ConA)-coated 96-well glass-bottom plate and incubated at 30°C for 1–2 hours prior to imaging. Images were acquired at 30°C at 30-second intervals over a 30-minute period following the addition of 150mM LiCl. For each replicate, three regions of interest were imaged. Bright-field images were captured at 30% intensity with a 40ms exposure, and GFP fluorescence images were acquired at 30% intensity, with a 300ms exposure. All imaging parameters were kept consistent across samples.

Cell segmentation and tracking was performed using the Cell-ACDC software [43] and embedded YeaZ cell segmentation algorithm [44]. Automated segmentation and tracking results were manually corrected frame by frame to ensure accuracy. Segmentation data were analyzed in Jupyter Notebooks using custom Python scripts. A localization ratio was calculated for each cell at each timepoint using the following formula:

$$\text{Localization ratio} = \frac{(\boldsymbol{q}95 - \textbf{median}) - (\textbf{median} - \boldsymbol{q}05)}{(\boldsymbol{q}75 - \boldsymbol{q}25)}$$

where q05, q25, q75, and q95 represent the 5th, 25th, 75th, and 95th percentiles of GFP pixel intensity, respectively. This formula is derived from Bowley's skewness [45] but is modified to measure tail asymmetry around the median. Positive values indicate the pixel distribution for a specific cell is skewed and has a long upper tail, consistent with nuclear localization of the reporter. The localization ratio is an effective proxy for nuclear signal.

## Hcm1 activity assays

Hcm1 activity was measured by expressing different Hcm1 proteins in an *hcm1Δ* reporter strain before or after exposure to stress. At the time of collection, strains were resuspended in sodium citrate buffer (50mM sodium citrate, 0.02% NaN$_3$, pH 7.4) and GFP levels in each cell were quantified using a Guava EasyCyte HT flow cytometer and GuavaSoft software. 5000 cells were measured in all samples. Mean fluorescence activity (MFI) was calculated using FlowJo software. Additional aliquots of cells were saved for analysis of cell cycle position by measuring DNA content, and for analysis of Hcm1 and GFP levels by Western blot.

## RNA purification

Cells amounting to five optical densities were harvested, pelleted by centrifugation at 3000rpm for three minutes, and stored at -80°C. Cell pellets were then thawed on ice, resuspended in 400µL AE buffer (50mM sodium acetate pH 5.3, 10mM EDTA), and moved to room temperature. 40µL 10% SDS and 400µL AE equilibrated phenol was added to each sample and thoroughly mixed by vortexing for 30 seconds. Samples were heated to 65°C for eight minutes and frozen in a dry ice and ethanol bath for five minutes. Organic and aqueous layers were separated by centrifugation at max speed for eight minutes at room temperature. The aqueous layer was then transferred to a new tube. To remove any residual phenol, 500µL phenol:chloroform:isoamyl alcohol was added and thoroughly mixed by vortexing for 30 seconds. Samples were incubated at room temperature for five minutes and the aqueous and organic layers were separated by

centrifugation at maximum speed for five minutes at room temperature. The aqueous layer was transferred to a new tube (~450µL) and the nucleic acids were precipitated by adding 40µL 3M NaOAc pH 5.2 and 1mL 100% ethanol. Samples were mixed by vortexing for 15 seconds and frozen in a dry ice and ethanol bath until completely frozen. Samples were then centrifuged at maximum speed for 10 minutes at 4°C. Supernatants were decanted and pellets washed with 80% ethanol and centrifuged at maximum speed for two minutes at 4°C. Supernatants were removed, pellets allowed to dry completely and resuspended in 50µL water. DNA was degraded by treatment with DNaseI. Samples were transferred to PCR strip tubes, 10µL 10X DNaseI buffer, 2µL DNaseI and 38µL water was added to each sample, and the samples were mixed by vortexing. Samples were incubated at 30°C for 30 minutes, then cooled to 4°C in a thermocycler. 1µL 0.5M EDTA was added to each sample and mixed. Samples were then heated to 75°C for 10 minutes and cooled to 4°C in a thermocycler. Purified RNA (100µL) was then transferred to a new tube and precipitated by adding 10µL sodium acetate pH 5.2 and 250µL 100% ethanol, and frozen in a dry ice and ethanol bath until completely frozen. RNA was then pelleted by centrifugation at maximum speed for 15 minutes at 4°C. Supernatants were decanted, the pellets washed with 80% ethanol, centrifuged at maximum speed for 2 minutes at 4°C. Supernatants were decanted and pellets allowed to air dry. Purified RNA was resuspended in water. Three biological replicates were performed. Library preparation and sequencing, including polyA mRNA selection, strand specific library preparation, and paired-end 100 base pair sequencing, were performed by Innomics/BGI Americas. All sequencing data is available in NCBI GEO and is accessible through GEO accession number GSE276435.

### RNAseq analysis

RNASeq analysis was performed with OneStopRNAseq [46]. Paired-end reads were aligned to Saccharomyces_cerevisiae.R64-1–1, with 2.7.7a [47], and annotated with Saccharomyces_cerevisiae.R64-1-1.90.gtf. Aligned exon fragments with mapping quality higher than 20 were counted toward gene expression with featureCounts [48]. Differential expression (DE) analysis was performed with DESeq2 [49]. Within DE analysis, 'ashr' was used to create log2 Fold Change (LFC) shrinkage [50] for all possible comparisons of WT and mutant strains, in both control and LiCl conditions. Significant DE genes (DEGs) were filtered with the criteria FDR<0.05. Gene set enrichment analysis was performed for Hcm1 targets genes using GSEA [51] on the ranked LFC. LFC analyses and GSEA results are included in S3 Dataset.

### Analysis of cell cycle position by flow cytometry

To analyze DNA content by flow cytometry, cells amounting to 0.15 optical densities were collected, fixed in 70% ethanol and stored at 4°C. Cells were then pelleted by centrifugation at 3000rpm for three minutes, resuspended in 1mL sodium citrate buffer (50mM sodium citrate, 0.02% NaN$_3$, pH 7.4) and sonicated. Samples were then pelleted by centrifugation, resuspended in 1mL sodium citrate buffer containing 0.25mg/mL RNaseA, and incubated at 50°C for one hour. 12.5µL 10mg/mL Proteinase K was added to each tube and samples were incubated for an additional hour at 50°C. Following incubation, 1mL sodium citrate buffer containing 0.4µL Sytox green was added to each sample and samples were left at room temperature for 1 hour or 4°C overnight, protected from light, for staining. DNA content was analyzed on a Guava EasyCyte HT flow cytometer and GuavaSoft software. 5000 cells were measured in all samples. Results were analyzed using FlowJo software. Percent cell cycle progression was calculated using the following equation: % progression = ((H − 1C)/(2C-1C)) × 100, where H = histogram mean fluorescence intensity (MFI), 1C = MFI of 1C DNA content peak, and 2C = MFI of 2C DNA content peak.

### Supporting information

**S1 Fig.  Controls for fitness experiments and Phosphosite Scanning.** (A) Schematic of pairwise competitive fitness assays. (B-C) GFP markers do not influence fitness. WT (*HCM1*) strains expressing GFP (GFP+) or GFP-Y66F (GFP-) were co-cultured in control medium (B) or medium containing LiCl (C). Strains have equivalent fitness. (D-E) *HCM1* and

*hcm1-8E* strains were grown in monoculture following the dilution protocol used for Phosphosite Scanning. (D) DNA content as measure by flow cytometry, which is used to infer cell cycle position. (E) Western blots showing Hcm1 expression levels. WT Hcm1 that is expressed from the genome under the control of the *GAL1* promoted was detected with HA-antibodies and was undetectable after 16 hours of growth in dextrose (dex). Plasmid expressed copies of Hcm1 are tagged with V5 and were expressed throughout the experiment. In control medium, levels of Hcm1 decrease at 24 and 48 hours when cells reach saturation but increase again as cells resume cycling. Hcm1-V5 expressed from the genomic locus is included for comparison, demonstrating that the plasmid expressed copy of Hcm1 is not overexpressed. PSTAIRE is shown as a loading control.
(TIFF)

**S2 Fig. Supporting data for Phosphosite Scanning screens.** (A-C) Scatter plots comparing selection coefficients derived from Hcm1 Phosphosite Scanning screens carried out with dilutions every 12-hour time points, to maintain logarithmic growth, (from [20]) to those carried out with dilution every 24-hours. (A) shows screening of the Hcm1 A/E library, (B) shows screening of the Hcm1 WT/A library, (C) shows screening of the Hcm1 WT/E library. (D-F) Heat maps showing log2 fold change values of each mutant in Phosphosite Scanning screens performed in control medium or LiCl (L). Each row represents a mutant, shown is the log2 fold change of normalized read counts with respect to time zero for each mutant, all mutants have been normalized to WT. Blue indicates depletion, red indicates enrichment. Mutants are clustered by the number of phosphomimetic mutations and ranked by increasing log2 FC within each cluster in control conditions. Shown is an average of n = 4 biological replicates.
(TIFF)

**S3 Fig. Dynamics of CN activation by LiCl.** (A) Representative image of cells expressing a GFP-tagged CN-reporter (CNR) that localizes to the nucleus upon CN activation. Cells are shown before and after treatment with 150mM LiCl for 3 minutes. Scale bar represents 10µm. (B) Active CN is detected in WT cells through 72 hours of LiCl treatment. WT (*CNB1*) and CN-mutant (*cnb1Δ*) cells expressing CNR were treated with 150mM LiCl and imaged. Shown is the percentage of cells with nuclear-localized reporter at the indicated time points. Data is an average of n = 3 experiments, error bars represent standard deviation. (C) Maximal CN activation peaks at approximately 5 minutes after LiCl addition. WT cells expressing the CN reporter were treated as in (A) and imaged at 30 second intervals for 30 minutes. Average localization ratio from n = 405 cells is shown, shaded area represents 95% confidence interval. (D) Representative traces of 4 cells from experiment in (C) showing short pulses of CN activation occur after the initial nuclear signal diminishes.
(TIFF)

**S4 Fig. Controls for Hcm1 reporter assays.** (A) FACS plots showing the distribution of GFP expression levels in cells expressing the indicated Hcm1 mutants or an empty vector (EV), after 40 hours of growth in control medium or medium with LiCl. Shown is a representative experiment from one of n = 6 replicates included in Fig 7A. (B) Western blots showing expression of GFP, Hcm1 mutants, and G6PDH (loading control) from a representative experiment from one of n = 6 included in Fig 7A. (C) FACS analysis of mutants included in Fig 7A after 40 hours of growth in control of LiCl medium. Representative plots from one of n = 6 experiments are shown on the left. A graph of the average percentage of cells in S-phase from all replicates is shown at the right. Error bars represent standard deviation. Significance was calculated comparing the percent S-phase in cultures in control vs LiCl medium using two-way ANOVA with Šídák's multiple comparisons test was used to test significance, *p < 0.01.
(TIFF)

**S5 Fig. Characterization of Hcm1 mutant strains growing in stress.** (A) WT and Hcm1 mutant cells do not release synchronously from a G1 arrest after they have adapted to LiCl stress. Monocultures of WT cells, or cells with the indicated mutants integrated into the *HCM1* genomic locus, were grown in control medium or LiCl medium for 40 hours. Cells

were then arrested in G1 with alpha-factor for 3 hours, released from the alpha-factor arrest, and cell cycle position was followed by flow cytometry. LiCl was kept in the medium in the indicated samples throughout the experiment. Representative plots for WT cells are shown on the left, quantitation of progression through the cell cycle is shown on the right. An average of $n = 3$ experiments is shown, and error bars represent standard deviations. Note that all strains growing in LiCl display a similar asynchronous release and are not significantly different from one another (as determined by two-way ANOVA with Geisser-Greenhouse correction and a Dunnett's multiple comparisons test). (B) Representative FACS plots showing the DNA content of the indicated Hcm1 strains after 40 hours of growth in control or LiCl containing media (left) and percentage of S-phase cells (right). Percentage of S-phase cells is an average of $n = 3$ biological replicates, error bars represent standard deviations. One-way ANOVA with the Geisser-Greenhouse correction and Tukey's multiple comparison test showed that none of the samples were significantly different from each other (ns).
(TIFF)

**S1 Table. Strain table.** All *S. cerevisiae* strains are in the BY4741 background.
(PDF)

**S2 Table. Plasmid table.** Hcm1 expression plasmids.
(PDF)

**S3 Table. Oligonucleotide table.** Sequences of oligonucleotides used for Phosphosite Scanning sequencing library construction.
(PDF)

**S1 Dataset. Selection coefficient values for Phosphosite Scanning screens.**
(XLSX)

**S2 Dataset. Statistical analyses of Phosphosite Scanning Data.**
(XLSX)

**S3 Dataset. Analyses of RNAseq experiments.**
(XLSX)

**S4 Dataset. Data underlying graphs in the paper.**
(XLSX)

## Acknowledgments

The authors thank Tom Fazzio and members of the Benanti lab for insightful discussions and critical reading of the manuscript, and Rob Brewster for assistance with imaging data analysis.

## Author contributions

**Conceptualization:** Michelle M. Conti, Jennifer A. Benanti.

**Formal analysis:** Michelle M. Conti, Jillian P. Bail, Aurelia R. Reynolds, Linnea G. Budge, Rui Li, Jennifer A. Benanti.

**Funding acquisition:** Jennifer A. Benanti.

**Investigation:** Michelle M. Conti, Jillian P. Bail, Aurelia R. Reynolds, Linnea G. Budge, Mackenzie J. Flynn.

**Methodology:** Michelle M. Conti, Jennifer A. Benanti.

**Supervision:** Lihua Julie Zhu, Jennifer A. Benanti.

**Writing – original draft:** Michelle M. Conti, Jennifer A. Benanti.

**Writing – review & editing:** Michelle M. Conti, Jillian P. Bail, Aurelia R. Reynolds, Linnea G. Budge, Mackenzie J. Flynn, Rui Li, Lihua Julie Zhu, Jennifer A. Benanti.

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
