## [Decision Letter · Decision Letter 0]

9 May 2025

PGENETICS-D-25-00426

Dynamic phosphorylation of Hcm1 promotes fitness in chronic stress

PLOS Genetics

Dear Dr. Benanti

Thank you for submitting your manuscript to PLOS Genetics. After careful consideration, we feel that it has merit but does not fully meet PLOS Genetics's publication criteria as it currently stands. Therefore, we invite you to submit a revised version of the manuscript that addresses the points raised during the review process.

Please submit your revised manuscript within 60 days. If you will need more time than this to complete your revisions, please reply to this message or contact the journal office at plosgenetics@plos.org. Please include the following items when submitting your revised manuscript:

We look forward to receiving your revised manuscript.

Kind regards,

Amy E. Ikui

Academic Editor

PLOS Genetics

Monica Colaiácovo

Section Editor

PLOS Genetics

Aimée Dudley

Editor-in-Chief

PLOS Genetics

Anne Goriely

Editor-in-Chief

PLOS Genetics

**Additional Editor Comments:**

Your manuscript was reviewed by two experts in the field. Both reviewers expressed concerns that there are insufficient evidence to support the role of Cks1-dependent priming sites and the pulsatile nature of Hcm1 activation. Reviewer 1 suggested performing phosphosite scanning experiments using TP-to-SP mutants, which would permit phosphorylation while preventing Cks1 binding. To assess whether calcineurin-mediated regulation of Hcm1 enhances stress fitness, they also proposed generating an Hcm1 PxIxIT motif mutant to test for fitness loss under stress conditions. Reviewer 2 also noted that some conclusions are indirect and not fully supported, particularly regarding Cks1-dependent priming and Hcm1 pulse phosphorylation. Both reviewers also raised several minor concerns. Therefore, we expect authors to resubmit the manuscript after "major revision" in 60 days (an extension can be granted if needed).

**Journal Requirements:**

At this stage, the following Authors/Authors require contributions: Michelle M Conti, Jillian P Bail, Aurelia R Reynolds, Linnea G Budge, Mackenzie J Flynn, Rui Li, Lihua Julie Zhu, and Jennifer A Benanti. Please ensure that the full contributions of each author are acknowledged in the "Add/Edit/Remove Authors" section of our submission form.

The list of CRediT author contributions may be found here: https://journals.plos.org/plosgenetics/s/authorship#loc-author-contributions

https://journals.plos.org/plosgenetics/s/submission-guidelines#loc-parts-of-a-submission

5) We have noticed that you have uploaded Supporting Information files, but you have not included a list of legends. Please add a full list of legends for your Supporting Information files after the references list.

Potential Copyright Issues:

- Figure 2. Please confirm whether you drew the images / clip-art within the figure panels by hand. If you did not draw the images, please provide (a) a link to the source of the images or icons and their license / terms of use; or (b) written permission from the copyright holder to publish the images or icons under our CC BY 4.0 license. Alternatively, you may replace the images with open source alternatives. See these open source resources you may use to replace images / clip-art:

**Reviewers' comments:**

Reviewer's Responses to Questions

**Comments to the Authors:**

Reviewer #1: In this report, Conti et al. investigate how dynamic phosphorylation of a yeast transcription factor, Hcm1, provides a fitness advantage under chronic stress conditions. They show that Hcm1 is rapidly dephosphorylated in response to LiCl stress and that constitutive activation by phosphomimetic mutations imparts a fitness disadvantage. They use phosphosite scanning to assess how phospho-null and phosphomimetic mutations in the Hcm1 transactivation domain (TAD) affect cellular fitness. This shows that nearly all mutations impart a reduced fitness advantage under stress conditions relative to non-stress controls. The authors attribute this to the disruption of phosphorylation dynamics, as opposed to simple changes in total activity or phosphorylation levels. To support this model, they use stabilizing mutations to show that increasing Hcm1 activity can provide a fitness advantage under all conditions, but only when TAD phosphorylation can also be dynamically regulated. Phosphosite scanning also allowed them to identify two putative CKS1 priming sites, in addition to the two that have already been established. Beyond the effects on fitness, they show that mutations that prevent dynamic phosphorylation disrupt Hcm1 transcriptional outputs during chronic stress.

This work provides compelling evidence that dynamic phosphorylation regulates Hcm1 signaling and is necessary for optimal fitness under stress. However, the narrative is difficult to follow, and we suggest it be restructured before publication. Additionally, this work would be substantially strengthened with experimental evidence to tie the effects on fitness they observe to the molecular mechanisms they describe. Overall, this work addresses underappreciated and difficult to study component of signaling networks. It represents a significant step forward in the field and we support its publication with the following comments and suggestions:

1. The EE mutation is described as a separation of function, isolating CKS1 dependent effects from all others. However, phosphomimetic mutations may not fully recapitulate other components of phosphorylation function. Phosphosite scanning experiments using TP to SP mutants, which would facilitate phosphorylation while still preventing CKS1 binding, would help to show defects in CKS1 priming are responsible for the observed reduction in fitness.

2. This story focuses on phosphorylation dynamics but does not directly address the phosphatase, calcineurin. If calcineurin is the only phosphatase targeting Hcm1, preventing interactions between it and Hcm1 (by mutating the Hcm1 PxIxIT motif) would result in constitutively hyper-phosphorylated Hcm1. If dynamic phosphorylation is required for optimal fitness, then preventing calcineurin-mediated regulation should result in a fitness loss under stress when compared to non-stress conditions.

3. Figure S3B shows Hcm1-8E and Hcm1-3N8E have reduced expression under LiCl treatment. Why is this the case and could it intersect the effects on survival the author’s report?

4. In figure 1D, it’s not clear that the growth disadvantage brought on by hcm1-8e is consistent over time. Why is this the case? Do cells adapt to higher activity over time?

5. Beginning at line 116, it is stated that cells were diluted every 24 hours to allow them to exit the cell cycle. Why is this a benefit?

6. In the third paragraph of the introduction, beginning at line 36, the authors write that “stressors signal” and “coordinate the stress response program”. This reads as though the stressors exhibit control over the cell as opposed to the cell responding to the stressor, which is likely more appropriate.

7. Figure panel ordering is inconsistent and hard to follow. In particular, figure 4 needs attention.

8. The title of the section on line 244 “Frequency modulation enhances Hcm1 activity in stress” does not seem appropriate. The associated experiments do not address how signaling amplitude or frequency impact Hcm1 signaling.

9. Figures S1B and S1C do not appear to be labeled correctly. Both grey and black points are labeled only as HCM1.

10. Throughout the text, there are small typos that need attention. For example, on line 318, "Csk1" should be corrected to "Cks1." Additionally, it would be helpful to maintain consistency with phosphorylation site nomenclature by consistently using either "T" (Threonine) or "S" (Serine) in front of the residue number.

Reviewer #2: Review of “Dynamic phosphorylation of Hcm1 promotes fitness . . . “, by Conti . . . Benanti, for PLoS Genetics

The authors make various mutations in the HCM1 gene that interfere with phospho-regulation of the protein, and measure the fitness of these mutants under normal and stressed conditions. They find that phospho-regulation contributes to fitness.

I think the main issue with the manuscript is that it wasn’t that interesting. I felt it was a long hard read, with small effects, and in the end, I didn’t learn that much.

PLoS Genetics lists some criteria for publication, and I’ll go through these.

Originality. The work is reasonably original. Not highly original—mutating phosphorylation sites and assaying phenotypes is a standard thing. But there are original aspects here, doing it with Hcm1 and in many combinations and measuring fitness.

High importance to researchers in the field. My feeling is no, it isn’t that important to workers in the field. The eventual conclusion is more-or-less what one would have said, that phosphorylation is for dynamic control.

Broad interest. Again, my feeling is no. This is not really a topic of broad interest.

Rigorous methodology. Yes. A tremendous amount of good, hard work went into this manuscript, there are a lot of data, and the data look very reliable.

Substantial evidence for its conclusions. Yes and no. The data are very good, and they show what they show. On the other hand, some of the conclusions—the inferences—are indirect, and are not conclusive. I am thinking for instance of the inferences about priming with Cks1, and about the need for pulses of Hcm1 phosphorylation. I thought the arguments for these inferences were reasonable, but still far from conclusive. “Substantial”, I’m not sure.

Some scientific points:

The disadvantage of hcm1-8E cells in stress. Lines 92 to 94 say that hcm1-8E cells have a fitness defect in stress (Fig. 1D). This finding is sort of what the paper is all about. But then, to the contrary, lines 131 to 133, the same mutant “had a slight fitness advantage in pooled screens (Fig. 2D . . .” So, the thing the paper is about—the mutant’s fitness disadvantage—isn’t even true when the authors do the assay a different way. To their credit, they recognize this and point it out, but even so, it’s an issue. Of this issue, they write: “This observation is consistent with previous findings that pooled screens result in higher selection coefficients than pairwise competition assays and is likely due to technical differences in the experimental approach . . .” Well, ok, but this doesn’t really explain anything to me, or make me feel better about the opposite results.

Also on this subject, I have to wonder a bit about the math of the selection coefficient, which is what they’re measuring in these assays. In a pairwise comparison, the math is straightforward, and they describe it in Methods, and that is fine. But in a pooled comparison, it seems to me more complicated, and I didn’t see a detailed description of what calculation they actually perform. So, for instance, suppose we have a pairwise comparison of strains A and B. Selection coefficient is straightforward. But now suppose we have A, B, and C (the pooled strategy). Suppose C does best and expands in the population, B does second best, but contracts, and A does worst and contracts a lot. Now, by the usual convention, B has a negative selection coefficient. But, it does better than A. So what are they reporting here? The absolute coefficient? I guess not, but they don’t say. Or the relative coefficient, relative to WT? I think so. But now I’m not 100% sure that the math is still the obvious math.

B. Cks1 priming. There was a lot of text about Cks 1 priming. On lines 174-175, they write “. . . which is consistent with Hcm1 regulation by Cks.” And that is right, it is consistent. But being “consistent” is a long, long way from proof that there is an important Cks1 priming effect. They do have other arguments, but all of them seemed to me indirect and inconclusive. And, mostly, the Ser mutations looked a lot like the Thr mutations, maybe arguing (gently) that the effects are not priming effects. So here, I thought the authors were probably right, but it is a long way from conclusive, and that is usually not good enough for publication.

One could ask, what experiments might support a role for Cks1 here? And there are experiments. But they involve doing a lot more long hard work, and I don’t see that ultimately it matters all that much anyway whether there is Cks1 priming or not, and so I do not want to recommend that any of these experiments be done.

C. Pulses of Hcm1 activity. The authors argue towards the end of Results that pulses of Hcm1 activity are important. The phosphatase calcineurin (CN) is unusual, in that it has rapid (minutes? I forget) cycles of on/off activity, which lead to pulses in the activity of a client transcription factor, Crz1. The authors argue that there might also be similar rapid pulses in activity of Hcm1, and these pulses might be important (and this would be quite a novel thing.). However, first, the authors have no direct evidence for any pulsing in Hcm1 activity. Second, even if there is “pulsing”, it could be on the cell-cycle time-scale (which would be an ordinary, non-novel possibility, since Hcm1 is a cell cycle transcription factor) rather than a Crz1-like minutes (?) timescale.

Here, it might be nice to do an experiment using a stressor that doesn’t work through calcineurin (say, high growth temperature, or an ethanol carbon source), and see if the hcm1 mutations have the same effect as when the stressor works through calcineurin. This gets at the pulsatile nature of calcineurin, and also uses stressors that are maybe more often relevant for yeast.

Ultimately, the author’s main conclusion—that dynamic phosphorylation of Hcm1 contributes to fitness under stress—is well-supported and convincing. But it is well-established that, in general, phosphorylation is for situations where dynamic regulation is needed, so it is hard to see this as a big step forward. Also, the "dynamic" phosphorylation could just be garden-variety cell cycle phosphorylation. Some of the other conclusions—about Cks1 priming, and the possibly pulsatile nature of Hcm1 activation—were not so well supported.

**Have all data underlying the figures and results presented in the manuscript been provided?**

Reviewer #1: Yes

Reviewer #2: Yes

PLOS authors have the option to publish the peer review history of their article (what does this mean? ). If published, this will include your full peer review and any attached files.

**Do you want your identity to be public for this peer review?** For information about this choice, including consent withdrawal, please see our Privacy Policy .

Reviewer #1: No

Reviewer #2: No

**Figure resubmission:**
---

## [Decision Letter · Decision Letter 1]

5 Sep 2025

Dear Dr. Benanti

We are pleased to inform you that your manuscript entitled "Dynamic phosphorylation of Hcm1 promotes fitness in chronic stress" has been editorially accepted for publication in PLOS Genetics. Congratulations!

Yours sincerely,

Amy E. Ikui

Academic Editor

PLOS Genetics

Monica Colaiácovo

Section Editor

PLOS Genetics

Aimée Dudley

Editor-in-Chief

PLOS Genetics

Anne Goriely

Editor-in-Chief

PLOS Genetics

Comments from the reviewers (if applicable):

Reviewer #1:

Reviewer #2:

Reviewer's Responses to Questions

**Comments to the Authors:**

Reviewer #1: We thank the authors for addressing our questions and comments with new experimental data and discussions, and we fully support the publication of this manuscript. We have no additional experimental suggestions within the scope of this manuscript.

We acknowledge the authors' conclusion that signaling networks often consist of components that, when analyzed in isolation, can make data interpretation challenging. For example, substituting threonine with serine requires consideration of their distinct phosphorylation and dephosphorylation rates as phospho-acceptors or donors. Additionally, effectively inhibiting calcineurin activity can be challenging due to the degenerate nature of its docking motifs. A cryptic PxIxIT/LxVP motif could facilitate a small amount of calcineurin activity but remain undetectable with existing SLiM prediction tools.

Reviewer #2: Re-review of “Dynamic phosphorylation of Hcm1 promotes fitness in chronic stress”, by Conti, . . . and Benanti, for PLoS Genetics

The authors make various mutations in the HCM1 gene that interfere with phospho-regulation of the protein, and measure the fitness of these mutants under normal and stressed conditions. They find that phospho-regulation contributes to fitness.

I feel a little worse about the manuscript than I did before.

In general, a scientific paper has two kinds of things in it: data from well-defined experiments, and inferences/conclusions. What other scientists are really relying on is the first, the data. The inferences are maybe more interesting, but there is usually some degree of doubt about them, and I care less about the inferences and more about the data.

For this manuscript, I think there are a lot of data, and the data look quite reliable to me, and I would be happy to have the data published so that everyone has access to them. I don’t know that they have to be published in PLoS Genetics. For the inferences/conclusions, I think there is a high degree of uncertainty and doubt in this particular case. Although I would like to see the manuscript published somewhere, I am not highly supportive. Still, if the Editor believes it is appropriate for PLoS Genetics, then there is nothing wrong with that.

Both myself and Reviewer 1 had concerns that there was insufficient direct evidence for the role of Cks1-dependent priming sites, and for the importance of pulsatile, Hcm1 regulation by calcineurin. The authors understood these concerns, discussed them very reasonably and at length in their response, and did a few extra experiments. However, the results of the two experiments done in response to Reviewer 1 tended to worsen my concerns.

The first extra experiment is shown in the author’s letter as Fig. R1, and it shows the phosphorylation of Hcm1 when certain T residues are replaced by S residues. The results are (as the authors say) hard to interpret, but do not support a role for Cks1 priming. The authors try to interpret the results by saying that S residues are more easily phosphorylated than T residues, citing Ord . . . and Loog 2019. I looked at this paper, and they said the initial velocity of phosphorylation of S might be 2 to 3-fold higher than for T, but this is just initial velocity. I do not really think this explains Fig. R1, and even if it does, it casts the importance of Cks1 priming into doubt, since the T priming residue would, by this explanation, be a worse site than a non-priming S. If the manuscript were to be published, this Figure is perfectly good, relevant data, and should be shown.

The second experiment, a very good experiment suggested by Reviewer 1, is to delete the CN docking motif from Hcm1, and see if there is a fitness defect (Fig. R2). The authors did the experiment; there is a very clear result; there is no fitness defect. The authors are leaving it out of the paper. I strongly object to them leaving it out. It is a very good, very relevant experiment with a clear result, and I feel it is being excluded entirely because it tends to argue against the author’s model, which is not a good reason to exclude it. The authors argue that the experiment can’t be perfectly interpreted, which I guess is true, but many (most?) of the experiments that are in the manuscript are even harder to interpret.

To discuss this experiment a little more, the authors say it is hard to interpret because deleting the PSIEIQ docking motif does not completely prevent dephosphorylation. But, that is not really what they say in their previous paper analyzing the effect of this motif. There, Fig. 2 shows that deletion of PSIEIQ has a dramatic effect on the 2-hybrid association of calcineurin with Hcm1 (Fig. 2B), and also a dramatic effect on dephosphorylation (Fig. 2D, 2E), and the authors say “. . .we found that within 10 min of direct CN activation by the addition of CaCl2 to the growth medium, Hcm1 was dramatically dephosphorylated (Figure 2D). This dephosphorylation was blocked by deletion of the CN docking site in Hcm1 . . .” Note dephosphorylation is described as “blocked”, which seems clear and easy to interpret.

(Later, the authors qualify this a bit, at least for in vitro studies, by writing: “In addition, deletion of the PSIEIQ motif impaired Hcm1 dephosphorylation in vitro (Figure 3B and Supplemental Figure S1B), although it was not completely blocked. This partial dephosphorylation of the ΔPSIEIQ mutant suggests that this motif is less important to mediate the interaction between CN and Hcm1 in vitro than in vivo, where there are hundreds of other CN targets present that reduce the concentration of available CN.”)

Both experiments R1 (to a small extent) and R2 (to a large extent) undercut the author’s model, especially the idea that pulsatile activity of calcineurin is important. One might say, maybe the model is wrong. Or one might say, whether the model is right or wrong, anyway we would all like to see a complete set of experimental data, not biased by just picking the data favorable to the model.

In conclusion, three things.

1. Overall, I am not very supportive, but on the basis of the manuscript’s high-quality data, it would be OK to publish. Editor’s choice.

2. The very good, high-quality, high-relevance experiments shown in R1 and R2 must be included if the manuscript is published. I feel very strongly about this. These experiments suggest some doubt about some of the author’s conclusions, and that doubt is 100% appropriate.

3. The line in the abstract “Moreover, our data indicates that pulses of Hcm1 activity are necessary to maximize target gene expression in stress.”, and any similar lines appearing elsewhere, should be removed. I do not think this point was proven, and experiment R2 argues against it.

One other thing, as a note to the authors (and not as part of the review . . .). While looking at Hcm1 sequences, I noticed the sequence LLDPLPYSPLK, starting at about position 380. Compare this to the Cln2-Cdc28 “LP” docking sequences found in Peter Pryciak’s paper “Comprehensive analysis of G1 cyclin docking motif sequences that control CDK regulatory potency in vivo”, Bandyopadhyay . . . and Pryciak, Curr Biol. 2021. Note especially the CDK site found just downstream of the “LP” motif; this is like the motifs in Whi5 and Stb1.

**Have all data underlying the figures and results presented in the manuscript been provided?**

Reviewer #1: Yes

Reviewer #2: Yes

PLOS authors have the option to publish the peer review history of their article (what does this mean? ). If published, this will include your full peer review and any attached files.

**Do you want your identity to be public for this peer review?** For information about this choice, including consent withdrawal, please see our Privacy Policy .

Reviewer #1: **Yes: ** Mardo Koivomagi

Reviewer #2: No

**Data Deposition**

http://datadryad.org/submit?journalID=pgenetics&manu=PGENETICS-D-25-00426R1

**Press Queries**

---

## [Editor Report · Acceptance letter]

PGENETICS-D-25-00426R1

Dynamic phosphorylation of Hcm1 promotes fitness in chronic stress

Dear Dr Benanti,

We are pleased to inform you that your manuscript entitled "Dynamic phosphorylation of Hcm1 promotes fitness in chronic stress" has been formally accepted for publication in PLOS Genetics! Your manuscript is now with our production department and you will be notified of the publication date in due course.

With kind regards,

Zsofia Freund

PLOS Genetics

On behalf of:
